# Next-Generation HDAC Inhibitors: Advancing Zinc-Binding Group Design for Enhanced Cancer Therapy

**DOI:** 10.3390/cells14241997

**Published:** 2025-12-15

**Authors:** Mohammed Hawash

**Affiliations:** Department of Pharmaceutical Chemistry and Technology, Faculty of Pharmacy, An-Najah National University, Nablus 00433, Palestine; mohawash@najah.edu; Tel.: +970-2569939939

**Keywords:** histone deacetylases, zinc-binding groups, cells, hydroxamate, SAR, non-hydroxamate inhibitors, anticancer

## Abstract

**Highlights:**

**What are the main findings?**
This review systematically analyzes recently developed HDAC inhibitors in recent years, emphasizing structural evolution across CAP, linker, and zinc-binding groups (ZBGs).It highlights the emergence of hydroxamate and non-hydroxamate ZBGs.

**What are the implications of the main findings?**
Understanding SAR trends across diverse scaffolds provides a rational foundation for designing next-generation HDAC inhibitors with improved isoform selectivity, potency, and therapeutic tolerance.The integration of structural insights with updated clinical trial data supports more informed medicinal-chemistry strategies aimed at accelerating the development of clinically viable HDAC-targeted anticancer agents.

**Abstract:**

Histone deacetylases (HDACs) are pivotal epigenetic regulators that control gene expression, cell proliferation, and differentiation, and their dysregulation is closely associated with the onset and progression of multiple cancers. The therapeutic importance of these enzymes is reflected by FDA approval of HDAC inhibitors for oncology indications. Despite this clinical success, most FDA-approved agents employ conventional zinc-binding groups (ZBGs) such as hydroxamic acid and 2-aminoanilide, which are frequently linked to metabolic instability, genotoxicity, and poor pharmacokinetic behavior. These limitations have spurred the development of structurally diverse and safer HDAC inhibitors incorporating alternative ZBGs. This review provides a comprehensive analysis of recently developed HDAC inhibitors reported in the last few years, emphasizing their structure–activity relationships (SARs), chemical scaffolds, and binding features—including cap, linker, and ZBG motifs. Both hydroxamate-based and non-hydroxamate inhibitors, such as benzamides, hydrazides, and thiol-containing analogs, are critically evaluated. Moreover, the potency and selectivity profiles of these inhibitors are summarized across different cancer and normal cell lines, as well as specific HDAC isoforms, providing a clearer understanding of their therapeutic potential. Emerging dual-target HDAC inhibitors, such as HDAC–tubulin, HDAC–PI3K and HDAC–CDK hybrids, are also discussed for their synergistic anticancer effects.

## 1. Introduction

Although significant advancements in potential anticancer therapies have been made, cancer continues to be a leading global trigger of mortality [1,2]. The efficacy of current FDA-approved drugs is severely hampered by prevalent issues like multiple drug resistance and debilitating side effects, underscoring a critical and immediate need for novel, less toxic therapeutics [3,4]. Extensive ongoing research is dedicated to identifying compounds with improved safety profiles. In pursuit of this goal, drug designers are focusing on synthesizing new chemical structures that specifically inhibit key biological pathways central to cancer progression, such as EGFR, CDKs, Ras, HDACs, and tubulin proteins, which serve as foundational targets for innovative anticancer interventions [5,6,7].

In parallel with the discovery of new HDAC inhibitors, recent advances in HDAC-based therapeutic strategies have expanded the scope of cancer treatment beyond classical enzymatic inhibition [8]. Combination regimens pairing HDAC inhibitors with targeted agents—including kinase inhibitors, DNMT inhibitors, and immunomodulatory drugs—have demonstrated enhanced synergistic efficacy and are increasingly explored in clinical trials [9,10]. Moreover, the emergence of HDAC-directed PROTAC degraders represents a transformative approach aimed at selectively inducing the proteasomal degradation of HDAC isoforms rather than merely blocking their catalytic activity [11,12].

HDACs are essential enzymes that regulate gene expression by removing acetyl groups from histone proteins. This critical deacetylation process directly influences chromatin structure and, consequently, the accessibility of DNA for transcription. The human proteome contains 18 distinct HDAC isoforms, categorized into four main classes (I, II, III, and IV), each defined by specific biological functions and tissue distribution [13]. Dysregulation of HDAC activity is strongly implicated in various pathologies, including cancer, neurodegenerative, and inflammatory disorders [14]. In a cancer context, impaired HDAC function can promote cell proliferation and survival by repressing crucial tumor suppressor genes [15]. Given this significant role in disease pathogenesis, the pharmaceutical discovery and development of therapeutic HDACi have become a major focus of research in recent years [16,17,18].

While all four classes of HDACs have been implicated in various cancers, Class I HDACs (HDAC1, HDAC2, HDAC3, and HDAC8) are generally considered the most directly and frequently associated with tumorigenesis and are often the primary targets in initial cancer therapeutic strategies [19,20].

Over recent years, HDACi have garnered significant attention as promising anticancer therapeutics, demonstrating efficacy against both solid and hematological malignancies [21]. To date, regulatory authorities have approved four pan-HDACi; these approved agents, including the hydroxamate-based compounds Vorinostat (SAHA), Belinostat (PXD101), Panobinostat (LBH589), and the thiol prodrug Romidepsin (FK228), are a testament to the clinical viability of HDAC inhibition [22,23].

Structurally, HDACi are categorized according to their ZBGs, which fundamentally dictate their binding mode, potency, and selectivity. The major ZBG-based classes include hydroxamates, 2-aminobenzamides, cyclic peptides, thiols, and short-chain fatty acids. While hydroxamates and benzamides dominate clinically approved HDAC inhibitors, increasing concerns regarding their metabolic instability, off-target metal chelation, and toxicity have accelerated the search for safer, more pharmacologically favorable alternatives to next-generation HDACi [24,25]. The hydrazide group, which could be considered as next-generation agents, has recently gained significant attention as an emerging ZBG with promising medicinal chemistry attributes. As highlighted in the recent perspective by Raucci et al. [26], hydrazides offer improved chemical stability, reduced side effects, and a well-characterized therapeutic profile, supported by their presence in clinically established drugs such as isoniazid. Beyond their documented antioxidant, antimicrobial, and anti-inflammatory properties, hydrazides have demonstrated selective and potent HDAC1–3 inhibition, first evidenced by the high-throughput discovery of UF010. These growing findings collectively position the hydrazide moiety as a compelling next-generation ZBG, capable of overcoming key limitations associated with traditional HDAC inhibitors [27,28].

The ZBG represents the core pharmacophore responsible for coordinating with the catalytic Zn^2+^ ion, and even subtle structural modifications can significantly influence binding affinity, isoform selectivity, metabolic stability, and overall therapeutic performance [29]. Newly explored ZBGs (next generation)—such as hydrazides, thiol-based motifs, and other non-classical chelators—provide refined metal-binding geometries that improve stereoelectronic complementarity with individual HDAC isoforms while minimizing off-target metal chelation and toxicity [30,31]. These design innovations are central to next-generation HDACi, as they promote stronger zinc coordination, more favorable cap–linker–ZBG alignment within the catalytic channel, and enhanced pharmacokinetic behavior. Collectively, such optimized ZBG architectures serve as key drivers in the development of safer, more selective, and more potent next-generation HDAC inhibitors capable of delivering superior therapeutic outcomes in cancer therapy.

This review aims to provide a clear and comprehensive overview of the most recent and genuinely novel HDACi, with a specific emphasis on compounds reported in the past few years. In addition to summarizing hydroxamate- and non-hydroxamate-based HDACis, the review highlights structurally innovative scaffolds, detailed SAR features, isoform selectivity, and therapeutic potential. Particular attention is devoted to newly emerging HDACi with promising preclinical profiles that demonstrate clear potential for translation into future clinical applications, and FDA-approved agents. Moreover, a dedicated section discusses the chemical design and recent advances in dual-target HDAC inhibitors, an increasingly important strategy in modern anticancer drug development. By focusing on the most innovative agents and integrating updated insights from the latest literature, this review aims to enhance readability, address gaps highlighted by peer evaluation, and provide meaningful guidance for future medicinal chemistry efforts and the discovery of next-generation epigenetic therapeutics.

## 2. FDA-Approved and Clinically Approved HDACi

HDACi have emerged as one of the most promising classes of epigenetic modulators in cancer therapy. Based on their chemical scaffolds and ZBGs, HDACis can be broadly classified into hydroxamic acids, cyclic peptides, benzamides, and short-chain fatty acids, each exhibiting distinct selectivity profiles toward HDAC isoforms and diverse clinical indications [15,32].

Hydroxamic acid derivatives represent the most extensively studied class due to their strong chelation of the catalytic zinc ion in HDAC active sites [33]. Among them, Vorinostat and Belinostat were the first to achieve FDA approval (in 2006 and 2014, respectively) (Figure 1) for the treatment of cutaneous and peripheral T-cell lymphomas (CTCL and PTCL) [34,35].

The FDA’s approval of Duvyzat (givinostat) marks a major advancement in the treatment of Duchenne muscular dystrophy (DMD)—a severe genetic disease-causing progressive muscle weakness and degeneration. As a histone deacetylase (HDAC) inhibitor, Duvyzat works by modulating gene expression to enhance dystrophin production, reduce inflammation, and protect muscle tissue. Clinical trials confirmed its safety and efficacy, leading to its regulatory approval. While not a cure, Duvyzat represents a meaningful step forward, offering patients a targeted therapy that addresses key molecular mechanisms of DMD and bringing new hope for improved quality of life and future therapeutic innovation [45,46].

Other analogs, such as Panobinostat and Resminostat, have highlighted clinical efficacy in multiple myeloma (MM) and colorectal or hepatocellular carcinoma (HCC), respectively [9,47]. Resminostat is a hydroxamate-based HDACi with the chemical structure (E)-3-[1-[4-[(dimethylamino)methyl]phenyl]sulfonylpyrrol-3-yl]-N-hydroxyprop-2-enamide. Furthermore, this compound belongs to the class of hydroxamic acid derivatives and exerts its pharmacological activity through the classical hydroxamate–zinc ion chelation mechanism at the HDAC catalytic site [9,48]. The presence of the N-hydroxyacrylamide group serves as the zinc-binding pharmacophore (ZBG), while the pyrrole–sulfonyl–dimethylaminophenyl scaffold provides both hydrophobic and hydrogen-bonding interactions that contribute to HDAC selectivity and favorable pharmacokinetics [49].

Resminostat exhibits broad inhibitory activity against Class I, IIb, and IV HDAC isoforms, with notable selectivity toward HDAC1, HDAC3, and HDAC6. This dual inhibitory action allows it to modulate both nuclear histone acetylation and cytoplasmic deacetylation processes. Mechanistically, Resminostat induces cell cycle arrest and apoptosis by upregulating p21^Waf1/Cip1 and downregulating oncogenic transcription factors such as c-Myc. It also restores differentiation pathways and suppresses epithelial-to-mesenchymal transition (EMT), leading to decreased invasiveness in solid tumors [50,51].

Additionally, Resminostat exemplifies a well-balanced, broad-spectrum HDAC inhibitor with the capacity to modulate several oncogenic signaling pathways simultaneously. Its distinctive chemical scaffold—combining a pyrrole-based CAP group, optimized linker, and a hydroxamic acid ZBG—contributes to a pharmacological profile that supports both potency and tolerability. Importantly, Resminostat’s progression from preclinical models to clinical evaluation underscores its translational relevance, particularly in malignancies characterized by profound epigenetic dysregulation. The most compelling evidence has emerged in hepatocellular carcinoma and multiple myeloma, where Resminostat has demonstrated clinically meaningful activity, reinforcing the therapeutic value of multi-HDAC targeting approaches in complex, treatment-resistant cancers.

## 3. Zinc Chelation Mechanism Across Diverse HDAC Isoforms

The structural architecture of HDACi inhibitors follows a highly conserved pharmacophore model consisting of three essential domains: a surface-interacting cap (CAP), a channel-penetrating linker, and a metal-chelating ZBG [29]. The structural architecture of HDACi is characterized by a conserved pharmacophore model composed of three key elements: a CAP, a channel-penetrating linker, and a metal-chelating ZBG. While hydroxamic acids have traditionally served as the classical ZBG due to their strong affinity for zinc, their clinical translation is often limited by mutagenicity concerns and off-target toxicities [29,52]. Consequently, the design of next-generation HDACi increasingly emphasizes the development of alternative ZBGs capable of maintaining high potency while improving safety profiles. The in silico approaches—including virtual screening, docking, molecular dynamics, and predictive modeling—are becoming indispensable tools in the design of new or next-generation HDAC inhibitors. These methods accelerate the identification of promising scaffolds, clarify zinc-binding mechanisms at the molecular level, and support the optimization of potency, selectivity, and drug-like properties before synthesis. By integrating computational strategies with experimental validation, researchers can more efficiently advance HDACi candidates with enhanced safety and translational potential [53].

As illustrated in Figure 2, the CAP region typically contains an aromatic or heteroaromatic moiety that anchors the inhibitor at the rim of the HDAC active-site pocket, providing isoform-tuning opportunities through steric and electronic modifications. The linker segment, often a flexible aliphatic or semi-rigid aromatic chain, serves as the structural spine that positions the terminal ZBG deep within the catalytic channel [29,54]. Finally, the ZBG—shown in the figure as a classical hydroxamate—coordinates the catalytic Zn^2+^ ion and dictates both potency and isoform selectivity.

In the context of this review, the pharmacophore model illustrated in Figure 2 does more than depict the classical CAP–linker–ZBG arrangement; it provides a conceptual scaffold that allows for a critical evaluation of how recent HDAC inhibitor designs succeed—or fail—to overcome longstanding limitations in this drug class. By examining newly reported inhibitors through this structural lens, we highlight not only how modifications in each domain affect potency and HDAC isoform preference, but also how certain strategies address persistent challenges such as metabolic instability, off-target toxicity, and poor tumor selectivity. Notably, the growing shift toward non-hydroxamate ZBGs, including hydrazides, benzamides, and emerging heteroatom-based chelators, reflects an intentional move away from the well-known drawbacks of hydroxamic acids. This trend demonstrates a deeper medicinal chemistry rationale: enhancing pharmacokinetic stability and therapeutic tolerability without compromising zinc-binding efficiency. Thus, the figure serves as a visual framework that anchors our critical discussion of recent SAR data and underscores emerging design principles shaping next-generation HDAC inhibitors, particularly those aimed at achieving isoform selectivity, reducing systemic toxicity, and improving clinical translation potential.

The success of early-stage clinical candidates established the hydroxamate functional group as the dominant and most widely utilized ZBG for HDACi. As exemplified by the co-crystal structure of HDAC2 with Vorinostat (SAHA) (PDB: 4LXZ), Figure 3A, the hydroxamate moiety functions as a potent, bidentate ligand capable of chelating the catalytic Zn^2+^ ion within the HDAC active site with high affinity. This could illustrate the critical coordination chemistry: the two oxygen atoms of the hydroxamate group coordinate directly to the zinc ion, forming a stable five-membered chelate ring. This chelation effectively blocks the enzyme’s hydrolytic deacetylase activity by sequestering the Zn^2+^ cofactor, which is essential for activating the water molecule required for substrate turnover. This specific, high-affinity zinc chelation mechanism is central to the efficacy of seminal HDACi compounds such as Vorinostat, Belinostat, and Panobinostat [49,55].

The subsequent co-crystal structures of Vorinostat bound to the catalytic sites of other isoforms, such as HDAC6 (Figure 3B) and HDAC8 (Figure 3C), further confirm this conserved inhibitory principle. Despite the distinct cellular localizations and primary functions of these two enzyme classes—HDAC2 is a nuclear enzyme involved in gene silencing, while HDAC6 is predominantly cytoplasmic and targets non-histone substrates like alpha-tubulin, the core interaction remains structurally identical [56]. In both cases, the hydroxamate ZBG of Vorinostat acts as a potent, bidentate ligand that consistently chelates the essential catalytic Zn^2+^ ion. Furthermore, this chelation efficiently blocks the active site tunnel and prevents the zinc ion from initiating the hydrolysis of the natural acetyl-lysine substrate. The conservation of the deep, narrow active site pocket geometry across different HDAC classes is precisely what allows this structurally simple hydroxamate ZBG to achieve its characteristic broad-spectrum (pan-inhibitory) activity.

Finally, these clinical studies underscore the therapeutic potential and evolving diversity of HDACi. The progress across different chemical scaffolds highlights the ongoing efforts to balance potency, isoform selectivity, and safety—ultimately guiding the next generation of HDAC-targeted anticancer agents.

## 4. Hydroxamate-Based HDACi

The success of early-stage clinical candidates established the hydroxamate functional group as the dominant and most widely utilized ZBG for HDACi [30,57,58]. Hydroxamates function as highly potent, bidentate zinc-chelating ligands that anchor deeply within the HDAC catalytic pocket, forming a stable coordination complex with the catalytic Zn^2+^ ion and thereby efficiently suppressing enzymatic deacetylase activity. This robust zinc-binding capability underpins their presence in several first-generation HDAC inhibitors, including Vorinostat (SAHA), Belinostat, and Panobinostat, all of which leverage the strong, dual-oxygen coordination of the hydroxamate motif to achieve broad-spectrum HDAC inhibition. While this interaction confers substantial potency, it is also associated with well-recognized drawbacks—such as metabolic instability, off-target metal chelation, and dose-limiting toxicity—which collectively motivate the search for alternative ZBGs in next-generation HDAC inhibitor design [59,60]. Notably, compounds like Pracinostat and Abexinostat have advanced to Phase II or III clinical trials (Table 1), showing broader activity against solid tumors, leukemia, and lymphomas [61,62].

The clinical development landscape of hydroxamate-based HDAC inhibitors continues to mature, as evidenced by the advancement of multiple agents across early- and mid-stage clinical studies and the clear documentation of their progress through registered ClinicalTrials.gov identifiers (Table 1). Resminostat (4SC AG), for instance, has completed Phase II evaluation in cutaneous lymphomas (NCT02953301) [63,64], while Pracinostat (Helsinn Healthcare SA) has advanced through Phase II/III trials in acute myeloid leukemia (NCT03151408) [62,65,66]. Abexinostat, developed by Xynomic Pharmaceuticals, has likewise reached Phase I/II stages in non-Hodgkin lymphoma (NCT04024696) [35]. Recently developed agents such as Bisthianostat from Shanghai Therion Pharmaceutical (Phase I; NCT03618602) [67,68] and the pan-HDAC inhibitor Ivaltinostat (CrystalGenomics), which has undergone Phase I/II studies in pancreatic adenocarcinoma (NCT05249101), further highlight ongoing innovation within this class [69,70]. Additional inhibitors—including Quisinostat (Janssen; Phase Ib–IVa, NCT01486277) [71,72,73], CUDC-101 and CUDC-907 (Curis Inc.; Phase I, NCT01384799 and NCT01742988, respectively) [74,75,76,77,78,79], CHR-3996 (Chroma Therapeutics; Phase I, NCT00697879) [80,81], and MPT0E028 (Taipei Medical University; Phase I, NCT02350868)—have generated essential pharmacokinetic, tolerability, and preliminary efficacy data supporting their translational relevance [82,83]. More recently, REC-2282 (AR-42) from Recursion Pharmaceuticals has completed Phase II/III clinical evaluation in neurofibromatosis type 2 [84,85], while R-306465 (Johnson & Johnson; Phase I, NCT00677001) has demonstrated activity in advanced solid tumors [86,87]. The integration of these trial identifiers within the review enhances scientific transparency and underscores the growing clinical validation of hydroxamate-based HDAC inhibitors, collectively reflecting a pipeline that is both diversified and steadily advancing toward therapeutic application.

**Table 1 cells-14-01997-t001:** Representative Hydroxamate-Based HDACi in Clinical Trials: Structures, Company, Clinical Trials, HDAC Targets, and Cancer Applications.

Name and Structure	Company	Clinical Trial (ClinicalTrials ID)	Cancer Type	HDAC Class	Ref.
Resminostat 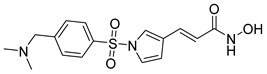	4SC AG	Phase II completed (NCT02953301)	Mycosis Fungoides, Sézary Syndrome, lymphoma	I & II	[63,64]
Pracinostat ^a^ 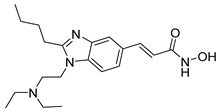	Helsinn Healthcare SA	Phase II/III; Ongoing/Completed (NCT03151408)	Acute Myeloid Leukemia	I, II,& IV	[62,65,66]
Abexinostat 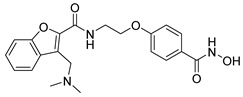	Xynomic Pharmaceuticals	Phase I/II; Completed/Ongoing (NCT04024696)	non-Hodgkin lymphoma	I & II	[35]
Bisthianostat 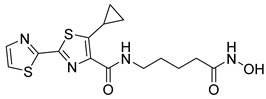	Shanghai Theorion Pharmaceutical	Phase I completed (NCT03618602)	Myeloma	pan-HDAC	[67,68]
Quisinostat 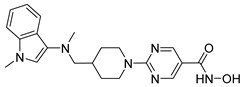	Janssen Research & Development	Phase Ib-IVa Completed (NCT01486277)	Cutaneous T-cell Lymphoma	I & II	[71,72,73]
Ivaltinostat ^b^ 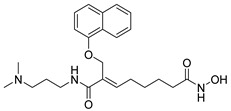	CrystalGenomics	Phase I/II Completed/Ongoing (NCT05249101)	Pancreatic Adenocarcinoma	pan-HDAC	[69,70]
CUDC-101 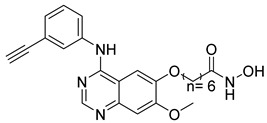	Curis, Inc.	Phase I completed (NCT01384799)	Head & Neck Cancer	I & II	[77,78,79]
CUDC-907 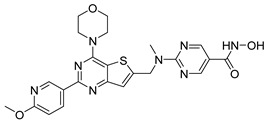	Curis, Inc.	Phase I completed (NCT01742988)	Lymphoma	I & II	[74,75,76]
CHR 3996 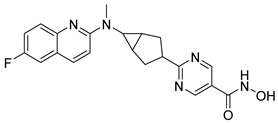	Chroma Therapeutics	Phase I completed (NCT00697879)	Solid tumors	I	[80,81]
MPT0E028 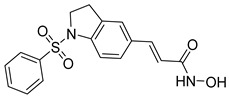	Taipei Medical University	Phase I completed (NCT02350868)	colorectal cancer and B-cell lymphoma	HDAC I, 2 & 6	[82,83]
REC-2282 (AR-42) 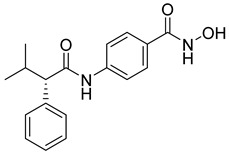	Recursion Pharmaceuticals	Phase II & III completed (NCT02350868)	Neurofibromatosis Type 2	HDAC I and IIb	[84,85]
R-306465 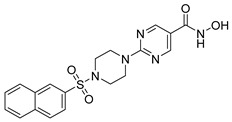	Johnson & Johnson Pharmaceutical	Phase I completed (NCT00677001)	Advanced Solid tumors	HDAC I	[86,87]

^a^ in Combination with Azacitidine; ^b^ in combination with Capecitabine or Capecitabine.

The hydroxamate-based HDACi included in this review share a conserved pharmacophore architecture consisting of a CAP group, a linker segment, and a ZBG. The hydroxamate moiety (–CONHOH), present across compounds such as Resminostat, Pracinostat, Bisthianostat, Ivaltinostat, and CUDC-series inhibitors, functions as a bidentate ZBG that strongly chelates the catalytic Zn^2+^ ion, facilitating broad and potent inhibition across HDAC classes I, II, and in some cases IIb and IV. The linker domain—typically a flexible aliphatic chain or an aromatic spacer—ensures proper alignment of the ZBG deep within the catalytic pocket while maintaining structural adaptability across various HDAC isoforms. Differences in linker length and rigidity, observed, for example, between the more extended linkers in CUDC-101 and CUDC-907 and the shorter, semi-rigid linkers in CHR-3996 and R-306465, correlate with differences in isoform preference and cytotoxic performance. The CAP region, which engages the rim of the active site, plays a dominant role in selectivity, cellular uptake, and pharmacokinetic behavior. Bulky aromatic CAP groups in agents such as Resminostat, Quisinostat, and CUDC-907 contribute to enhanced selectivity toward HDAC1/2/6 and improved interaction with surface residues, whereas heterocyclic CAP moieties found in CHR-3996 support tighter HDAC1-focused inhibition with reduced off-target effects. Likewise, the more voluminous CAP structures of pan-HDAC inhibitors such as Bisthianostat and Ivaltinostat enable broader engagement of class I and II enzymes, aligning with their wide-spectrum profiles in clinical evaluation. Overall, SAR trends across these inhibitors underscore the delicate interplay between CAP size and polarity, linker geometry, and hydroxamate ZBG positioning—each component collectively shaping HDAC isoform selectivity, potency, therapeutic window, and clinical translatability.

Many works were carried out by synthesizing compounds containing this scaffold with potent activities on HDAC isoenzymes [88,89,90]. Cursaro et al. designed and synthesized a series of vanillyl-based hydroxamic acid HDACi targeting HDAC6 and HDAC8, which are key epigenetic regulators implicated in aggressive neural tumors such as neuroblastoma and glioblastoma. SAR studies and molecular docking highlighted the influence of the CAP group (vanillyl), linker, and zinc-binding orientation on isoform selectivity. Among these synthesized derivatives, St.1 (Table 2) emerged as a highly potent and selective HDAC6 inhibitor (IC_50_ = 4.5 nM) [91].

In another study, a series of hydroxamic acid-based HDACi incorporating a tetrahydro-β-carboline core linked by an aliphatic chain was designed and synthesized. Among them, compound St.2 (Table 2) exhibited potent HDAC1 inhibition and strong antiproliferative activity across various tumor cell lines. Molecular docking studies supported its high binding affinity toward HDAC1, suggesting that this structural scaffold represents a promising framework for developing novel HDACi with potential efficacy against solid tumors [92]. In another study, two novel dihydroxamate-based HDAC1 inhibitors were designed and evaluated through molecular docking, molecular dynamics simulations, and enzymatic assays. Compound St.3 (Table 2) highlighted strong interactions with HDAC1, including hydrogen bonding and zinc coordination, and exhibited higher binding affinity (−6.2 kcal/mol), greater complex stability. These findings highlight St.3 as a promising lead for developing selective HDAC1-targeted anticancer agents [93]. In a recent study, a new series of N-hydroxycinnamamide-based hydroxamic acid derivatives was synthesized as potent HDACi targeting THP-1 monocytic leukemia cells. Among 22 compounds (including 20 novel ones), St.4 (Table 2) showed the strongest antiproliferative and pro-apoptotic effects, comparable to vorinostat, and effectively altered cell-cycle progression. Both compounds inhibited HDAC class I and II with similar potency to vorinostat and increased acetylation of histones H2A, H2B, H3, and H4. Molecular modeling confirmed zinc chelation by the hydroxamate group, supporting their potential as dual-class HDACi for leukemia therapy [94].

In another recent study, a novel hydroxamate-based HDAC1-selective inhibitor, compound St.5 (Table 2), was designed and synthesized to combat castration-resistant prostate cancer (CRPC). This compound exhibited remarkable HDAC1 selectivity, leading to increased histone H3 acetylation at Lys9/27 and potent cytotoxicity. Mechanistic studies revealed that St.5 induces G1 phase arrest through Cyclin D1 downregulation and triggers intrinsic apoptosis via caspase-3 activation and an elevated Bax/Bcl-2 ratio. Additionally, in vivo, St.5 significantly reduced tumor growth (by 75%) in PC3 xenograft models without evident toxicity, highlighting it as a promising selective HDAC1 inhibitor for advanced prostate cancer therapy [95]. In another study, a series of (arylidene)aminoxy-based HDACi were synthesized using the SAHA scaffold to explore their potential against uveal melanoma (UM), a highly aggressive and treatment-resistant cancer. The compounds were evaluated for inhibitory activity against HDAC1, 3, 6, and 8, supported by molecular docking to elucidate binding interactions. Among them, the quinoline derivative St.6 (Table 2) emerged as the most potent, showing nanomolar HDAC6 inhibition and notable antiproliferative activity on UM cell lines at micromolar concentrations. VS13 also modulated HDAC target gene expression comparably to SAHA, indicating its promise as a lead compound for developing HDAC6-targeted therapies in uveal melanoma [96]. In another investigation, a series of thiazole-based hydroxamate derivatives was developed as novel HDACi with strong anticancer potential. Among them, compound St.7 was the most potent against HDAC1 and showed significant cytotoxicity against HepG2 cells, accompanied by increased acetylation of histones H3 and H4. These findings identify thiazole-based hydroxamates as promising scaffolds for further development of multimodal HDAC-targeted anticancer agents [97]. In another study work, a series of tetrahydro-β-carboline-based hydroxamate derivatives were synthesized as novel HDACi. Among them, St.8 (Table 2) displayed potent HDAC1 inhibition and strong antiproliferative activity against the A549 lung cancer cell line. Treatment with St.8 led to elevated acetylation of histones H3 and H4, G2/M phase arrest, increased ROS generation, and DNA damage, ultimately triggering apoptosis [98].

In another work, nigranoic acid (NA) and manwuweizic acid (MA), two triterpenoids identified as HDACi via docking-based virtual screening, were chemically modified to generate a series of derivatives with enhanced biological activity. Among them, hydroxamic acid derivatives exhibited improved inhibition of HDAC1/2/4/6 (with the lowest IC_50_ = 1.14 μM) while showing no effect on HDAC8. Notably, compound St.9 (Table 2) highlighted strong anti-inflammatory activity in J774A.1 macrophages by increasing histone acetylation, inhibiting IL-1β maturation, caspase-1 cleavage, and NLRP3 inflammasome activation, without compromising cell viability. These findings highlight compound St.9 as a promising low-cytotoxic anti-inflammatory HDAC inhibitor and provide a valuable scaffold for probing the link between HDACs and inflammasome regulation [99].

A novel coumarin–hydroxamate hybrid (St.10; Table 2) was designed through a pharmacophore fusion strategy to enhance HDAC1 inhibition and overcome the limitations of current HDACi in breast cancer therapy. Molecular docking confirmed the strong binding affinity of St.10 to HDAC1, while in vitro and in vivo studies highlighted its potent anti-breast cancer activity with no systemic toxicity [100].

A study containing two hydroxamic acid derivatives, St.11 and St.12 Table 2, explored their potential as selective HDAC6 inhibitors for treating liver fibrosis. Chronic liver injury often progresses to fibrosis and cirrhosis due to persistent hepatitis, with limited effective therapies available. In this work, molecular docking, enzyme inhibition, and cellular assays were conducted to assess the specificity and anti-fibrotic effects of DR-3 and FDR2. Both compounds showed strong selectivity for HDAC6 over HDAC1, significantly reducing hepatic stellate cell (HSC) activation, fibrogenic gene expression, and collagen deposition [89].

In another work, a series of 3-substituted quinoline hydroxamic acids was synthesized and tested against A549 and HCT116 cancer cells and HDAC isoforms (1, 2, 6, 8). Substitution at C3 favored HDAC6 selectivity. Compound St.13 showed potent antiproliferative activity and nanomolar HDAC6 inhibition. Mechanistic studies revealed activation of caspase-dependent apoptosis, increased H2AX phosphorylation, and sub-G1 cell-cycle arrest, confirming HDAC6-targeted anticancer potential [101]. Recently, a series of imidazolyl-hydroxamic acid Schiff base derivatives was synthesized and evaluated as HDACi. Among them, St.14 (Table 2) showed the strongest anticancer activity across multiple cell lines, especially A2780 ovarian cancer, by inducing apoptosis, reducing colony formation and migration, and exhibiting antiangiogenic effects comparable to Vorinostat. In vivo, St.14 (50 mg/kg) outperformed Vorinostat in the 4T1 breast cancer mouse model. In silico studies confirmed stable HDAC–ligand interactions, suggesting St.14 as a potent lead for solid tumor therapy [102].

In another study, a novel series of 11 urushiol-based hydroxamic acid HDACi was designed and synthesized. These compounds displayed potent inhibition of class I HDACs (HDAC8, IC_50_ = 16–41 nM), with minimal activity against HDAC6. Docking studies highlighted key interactions contributing to HDAC8 inhibition, while Western blot analyses confirmed enhanced acetylation of histone H3 and SMC3, supporting selectivity for class I HDACs. St.15 Table 2 was the most active urushiol-based hydroxamates and represents promising scaffolds for further development as anticancer agents [103].

A new series of 2H-1,2,3-triazole-based hydroxamate analogs, inspired by vorinostat, was synthesized and evaluated for anticancer potential. Among 15 designed compounds, St.16 (Table 2) showed potent antiproliferative and cytotoxic effects against MCF-7 breast cancer cells with minimal toxicity toward normal HEK-293 cells. Mechanistic studies revealed apoptosis induction, S and G_2_/M phase arrest, and ROS generation. Molecular docking, HDAC inhibition assays, and MD simulations confirmed strong binding affinity toward HDAC1 and HDAC6 [104]. In another study, an HDAC6-selective inhibitor, St.17 (Table 2), was developed through a SAR approach for the treatment of T-cell prolymphocytic leukemia (T-PLL), an aggressive and currently incurable hematological malignancy. St.17 highlighted high potency and selectivity for HDAC6, with strong on-target activity and a favorable safety profile in non-transformed cells. In primary T-PLL patient cells, where HDAC6 is overexpressed, St.17 elicited robust biological responses, and combination studies revealed synergistic effects with approved anticancer agents, including bendamustine, idasanutlin, and venetoclax [105]. In another good work, a series of tetrahydro-β-carboline (THβC)-based hydroxamic acids was designed and synthesized as novel selective HDAC6 inhibitors using a scaffold-hopping strategy. Several analogues highlighted sub-nanomolar HDAC6 inhibition (IC_50_ < 5 nM) with strong selectivity. Molecular docking studies clarified that the SAR of St.18 (Table 2) was the most potent derivative and displayed favorable pharmacokinetic properties following oral administration in mice, highlighting its promise as a potent and selective HDAC6-targeted anticancer lead compound [106].

A series of estratriene-based hydroxamic acid derivatives was designed as HDACi using estrone and estradiol scaffolds linked via alkoxy chains. St.19 (Table 2) showed the best activity, exhibiting potent antiproliferative effects against HeLa. Molecular docking confirmed strong HDAC2 and HDAC6 binding through zinc coordination and favorable cap orientation, supporting these estratriene-based scaffolds as promising candidates for anticancer HDAC inhibitor development [107]. In another study, a series of hydroxamic acid-based HDAC6 inhibitors incorporating adamantane and natural terpene (camphane, fenchane) fragments as CAP groups were synthesized and evaluated through in silico, in vitro, and in vivo approaches. Among the synthesized compounds, St.20 showed the most potent HDAC6 inhibition and anti-β-amyloid aggregation properties, featuring an adamantane-amide-hydrocarbon scaffold, as a promising multitarget neuroprotective candidate for Alzheimer’s disease therapy [108].

Hydroxamate derivatives, including coumarin-, tetrahydro-β-carboline-, quinazolinone-, estratriene-, and urushiol-based scaffolds, have underscored potent and selective inhibition of various HDAC isoforms, especially HDAC1, HDAC6, and HDAC8, leading to cell cycle arrest, induction of apoptosis, and enhanced histone acetylation in multiple cancer models. SAR studies, supported by molecular docking and molecular dynamics simulations, have guided the optimization of zinc-binding groups, linkers, and surface recognition moieties to improve selectivity, potency, and pharmacokinetic profiles. In addition to classical cancer targets, dual- and multi-target inhibitors combining HDAC inhibition with PIM-1, PARP-1, IDO1, or ribonucleotide reductase inhibition have shown synergistic antiproliferative effects and promising in vivo efficacy.

## 5. Non-Hydroxamate-Based HDAC Inhibitors

HDACs have emerged as critical epigenetic targets in cancer therapy, neurodegenerative diseases, and immune-related disorders, prompting extensive research into both hydroxamate- and non-hydroxamate-based inhibitors. Non-hydroxamate HDACi, such as 2-aminobenzamides [109,110], cyclic peptides [111,112,113], thiols [114], coumarin-sulfonamide [115], and hydrazides [116], and other novel scaffolds, further expand the therapeutic potential by offering alternative mechanisms, reduced toxicity, and isoform-selectivity. Collectively, these studies highlight the versatility of HDAC-targeted small molecules as valuable frameworks for developing next-generation therapeutics with enhanced efficacy and safety profiles across diverse disease models [117]. Here, in this section, various works that do not contain a hydroxamate scaffold will be included accordingly.

Benzamide derivatives, including Entinostat, Chidamide, Tacedinaline, and Mocetinostat (Table 3), represent a newer generation of selective class I and class IIb HDACi [25,118]. These compounds generally possess improved safety and selectivity profiles, making them attractive for combination therapy. For instance, Entinostat and Chidamide have reached Phase III clinical trials for solid tumors and hematological malignancies; all clinically successful agents were listed in Table 3 with their chemical structures and targeted tumors. Structurally, these inhibitors share the canonical pharmacophore triad of CAP group, linker, and ZBG, which was presented before in Figure 2, but their variations in aromatic substituents and linker design strongly influence isoform selectivity and biological potency. All four compounds possess an ortho-aminobenzamide moiety as the ZBG, which coordinates with the catalytic zinc ion in a bidentate fashion while providing hydrogen bonding interactions within the enzyme pocket. This benzamide group binds less aggressively than hydroxamates, leading to enhanced selectivity toward HDAC1–3 and reduced off-target effects.

The CAP and linker regions are key determinants of activity and selectivity. Entinostat features a pyridine cap connected via a flexible carbamate linker, allowing optimal orientation at the rim of the catalytic site, while chidamide introduces fluorophenyl and pyridyl rings connected by a conjugated amide linker, improving rigidity, lipophilicity, and membrane permeability. Tacedinaline adopts a more compact diaryl amide framework, providing strong class I inhibition through a short, rigid linker that enhances zinc-pocket alignment. In contrast, mocetinostat incorporates heteroaromatic scaffolds in both the CAP and linker regions, improving hydrogen bonding and π-stacking interactions, thereby broadening its selectivity toward Class I and IV HDACs. Collectively, these structural variations illustrate the SAR principles governing benzamide HDACi—fine-tuning of cap polarity, linker rigidity, and ZBG geometry enables precise modulation of isoform selectivity, potency, and clinical performance.

A novel series of class I-selective HDACi containing 2-aminobenzamide zinc-binding groups linked to piperazinyl-pyrazine or piperazinyl-pyrimidine cores was synthesized and evaluated. Several compounds, including St.21 (Table 4), exhibited high selectivity for HDAC1, 2, and 3 over other HDAC isoforms and showed superior in vitro activity compared to clinically tested inhibitors such as Entinostat. Molecular docking and dynamics studies supported the observed structure–activity relationships [128]. In another amino-benzamide structure, the impact of incorporating a β-carboline cap into HDACi containing cinnamic acid linkers and benzamide zinc-binding groups was evaluated. A series of β-carboline–cinnamide conjugates was synthesized and evaluated for HDAC inhibition and antiproliferative activity against various human cancer cells. Most compounds exhibited superior HDAC inhibitory activity compared to the standard drug Entinostat, and St.22 (Table 4), showing notable potency against HCT-15 cells [129]. In a recent study, a series of novel benzamide derivatives with modified linker groups was synthesized as selective HDAC3 inhibitors. These compounds displayed potent antiproliferative activity against multiple cancer cell lines, with minimal toxicity toward normal human cells. Among them, St.23 (Table 4) showed significant HDAC3 selectivity (~47-fold over HDAC2), induced G_2_/M cell cycle arrest and apoptosis in 4T1 breast cancer cells, and underscored a favorable in vivo pharmacokinetic profile [130].

Selective HDAC6 inhibitors were designed using thiol as the ZBG to overcome the limitations of hydroxamate-based inhibitors, such as poor pharmacokinetics and potential genotoxicity. A series of thiol-based HDAC6 inhibitors was synthesized, and their SAR was analyzed via molecular docking. Notably, St.24 exhibited high selectivity (29-fold) while St.25 (Table 4) showed the highest activity against HDAC6. The use of a pyrimidine linker in these thiol-based inhibitors represents a novel scaffold, potentially offering improved pharmacokinetic properties and reduced genotoxicity compared to traditional hydroxamate derivatives [131]. In this study, selective Class I HDAC inhibition was explored using a cyclic peptide approach inspired by natural depsipeptides. Building on Largazole, a fluorinated analog of Romidepsin St.26 (Table 4) was synthesized in 12 steps. This analog showed potent inhibitory activity against Class I HDACs with negligible HDAC6 inhibition, confirming its high selectivity. This structure significantly inhibited the growth of NCI-H1975 and HT29 cancer cells while showing reduced cytotoxicity toward normal cell lines (WRL-68 and HEK293). Mechanistic studies underscored that St.26 induced cell-cycle arrest and apoptosis [132]. In recent work, 33 chrysin derivatives were synthesized and characterized to assess their HDAC inhibitory and anticancer activities. Among them, compound St.27 showed the strongest HDAC inhibition and was most selective toward HDAC8 [133].

In another work, a series of sulfur-based selective HDACi was developed by modifying ajoene, a natural compound from garlic, to overcome the toxicity issues associated with hydroxamic acid-based inhibitors. Structure–activity and docking studies revealed potent and highly selective HDAC8 inhibitors with a novel zinc-binding group. Among them, St.28 (Table 4) underscored strong antiproliferative activity against neuroblastoma cell lines and showed significant in vivo efficacy in a BE(2)-C xenograft mouse model, highlighting its promise as a new HDAC8-targeted anticancer agent [134].

A growing body of research has focused on developing non-hydroxamic HDACi to overcome the genotoxicity and poor pharmacokinetic limitations of hydroxamate-based agents [49]. Several alternative ZBGs have been explored, including thiol, carboxylate, sulfur-based, and aminobenzamide scaffolds, each offering enhanced selectivity and safety profiles. Thiol-based HDAC6 inhibitors, such as compound St.24, underscored notable potency and selectivity with potentially improved pharmacokinetics, while sulfur-containing derivatives like compound St.28 showed HDAC8 selectivity and in vivo antitumor efficacy in neuroblastoma models. Similarly, carboxyl-containing chrysin derivatives St.27 exhibited selective HDAC8 inhibition and antiproliferative activity against colon cancer cells. Among aminobenzamide analogs, several agents achieved class I or HDAC3 selectivity, strong apoptosis induction, and tumor growth suppression in vivo. Collectively, these studies highlight the promising therapeutic potential of non-hydroxamic HDACi with diverse ZBGs as safer, selective, and potent anticancer candidates.

## 6. HDAC-Based Dual-Target Inhibitors

Over the past five years, increasing attention from both academia and industry has been directed toward the development of isoform-selective HDACi and HDAC-based dual inhibitors (hydroximate and non-hydroximate-based scaffolds) that simultaneously modulate HDACs and complementary oncogenic pathways. This dual-targeting paradigm offers the potential to enhance therapeutic efficacy, reduce off-target effects, and overcome resistance mechanisms, leading to substantial progress and innovative designs in this rapidly evolving field [135,136]. A series of 3,4,5-trimethoxyphenyl-based hybrids was designed and synthesized by Mohamed et al. as dual EGFR/HDACi by combining pharmacophoric features of both targets. The derivatives containing hydroxamic acid showed the most potent anticancer activity against HepG2, MCF-7, HCT116, and A549 cell lines Table 5, and among them, compound St.29 exhibited remarkable EGFR inhibition (IC_50_ = 0.063 µM), comparable to staurosporine, and selective HDAC6 inhibition. Mechanistic studies confirmed its pro-apoptotic effect through increased Bax, caspase-3, and caspase-8 levels, along with reduced Bcl-2 expression and cell cycle arrest at G1/S phase [137]. In another work, a series of phenylurea hydroxamic acids was developed by combining the pharmacophores of HDAC and VEGFR-2 inhibitors. Among them, St.30 (Table 5) effectively inhibited HDAC and showed modest VEGFR-2 activity. Molecular docking revealed that the hydroxamic acid moiety maintains zinc coordination in HDAC while interacting with key residues in the VEGFR-2 active site [138].

In a related study, a new series of phthalazinone derivatives was designed and synthesized to target PARP-1 alone or both PARP-1 and HDAC1 simultaneously. Several derivatives displayed remarkable enzyme inhibition, with DLC-1 showing exceptional PARP-1 potency and strong antiproliferative effects across breast cancer cell lines, inducing G1 phase arrest and apoptosis. Among the dual inhibitors, St.31 (Table 5) exhibited potent activity against both PARP-1 and HDAC1. These results highlight phthalazinone-based hybrids as promising scaffolds for developing dual PARP-1/HDAC1 inhibitors with strong anticancer potential [139]. A series of dual PIM-1/HDACi was designed and synthesized based on a 3-cyanopyridine-hydroximate scaffold (zinc-binding groups). Several compounds exhibited broad-spectrum anticancer activity across the NCI-60 cancer cell line panel. Notably, hydroxamic acid St.32 (Table 5) showed potent antiproliferative effects and displayed strong activity (GI_50_ ≤ 3 μM) and inducing G_2_/M cell cycle arrest and pre-G1 apoptosis. Particularly, St.32, a promising dual PIM-1/HDACi with potential for further in vivo development in anticancer therapy [140].

In numerous studies, researchers have sought to design and synthesize diverse series of compounds exhibiting dual inhibitory activity against both HDAC and tubulin [6,141,142]. In another work on dual inhibitors, researchers developed a new class of dual tubulin/HDAC inhibitors derived from the natural product millepachine, a known tubulin polymerization inhibitor. Their study reported the synthesis and evaluation of several analogs, among which compound St.33 (Table 5) demonstrated exceptional potency against PC-3 prostate cancer cells, and effectively inhibited both microtubule polymerization and HDAC activity, leading to G2/M cell cycle arrest and pronounced apoptosis. In a PC-3 xenograft model, St.33 achieved a remarkable tumor inhibition rate of 90.07% at 20 mg/kg, significantly outperforming the reference compound CA-4. The compound also displayed favorable in vivo drug metabolism characteristics [141].

A series of phthalazino [1,2-b]-quinazolinone-based hybrids bearing ortho-aminoanilide or hydroxamic acid groups was developed as a multi-target HDACi for solid tumor therapy. Among them, St.34 showed nanomolar inhibitory potency against cancer cells and HDAC subtypes, surpassing SAHA (vorinostat). Mechanistic studies revealed that St.34 enhanced histone H3 and α-tubulin acetylation and activated the p53 signaling pathway, leading to HepG2 cell growth inhibition [143]. A 1,10-phenanthroline-based hydroxamate derivative, St.35 (Table 5), was synthesized accordingly and identified as a dual HDAC/RR inhibitor with notable anticancer potential. Dual inhibition induced ROS-mediated apoptosis, confirming synergistic action. Docking and molecular dynamics revealed strong binding to HDAC isoforms, especially HDAC7 (–9.63 kcal/mol), surpassing SAHA. These findings highlight St.35 as a promising dual-target anticancer scaffold for further development [144].

**Table 5 cells-14-01997-t005:** The structures of HDAC-based dual-target inhibitors and their IC_50_ values against a panel of cancer cell lines, HDAC enzymes, and other targets for the most active agents.

Code	Structures	Evaluated Cancer/Normal Cell Lines		Evaluated Target		Ref.
		Cell Lines	IC_50_/%V	HDAC	IC_50_	
St.29	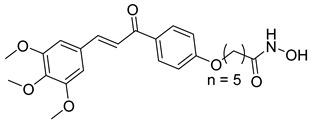	MCF-7 HepG2 HCT116 A549	0.621 µM 0.536 µM 1.206 µM 0.797 µM	HDAC1 HDAC2 HDAC4 HDAC6 HDAC8 EGFR	0.148 µM 0.168 µM 5.852 µM 0.060 µM 2.257 µM 63 nM	[137]
St.30	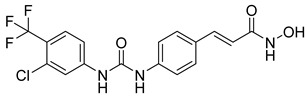	EPC	1.0 µM	HDAC6 EPCs	166 nM 1.0 µM	[138]
St.31	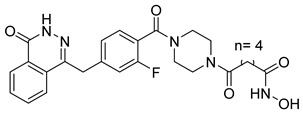	MDA-MB-436 MDA-MB-231 MCF-7	0.30 µM 2.70 µM 2.41 µM	HDAC1 PARP-1	31 nM <0.2 nM	[139]
St.32	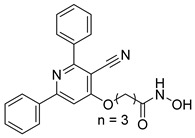	HeLa	65.94 nM	HDAC1 HDAC6 PIM-1	63.65 nM 62.39 nM 343.87 nM	[140]
St.33	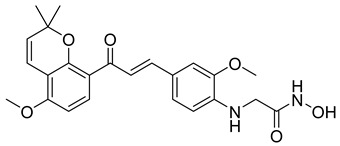	PC-3	16 nM	HDAC2 Tubulin	0.43 µM 4.82 µM	[141]
St.34	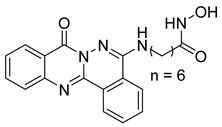	HepG2	0.58 µM	HDAC1 HDAC6	13.37 nM 42.74 nM	[143]
St.35	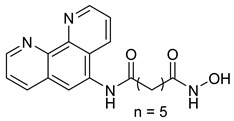	SiHa HepG2 MCF-7 Cal27	16.43 µM >100 µM >100 µM 50.98 µM	HDACs RR	10.80 µM 9.34 µM	[144]
St.36	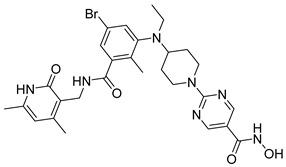	SU-DHL-6	1.20 µM	HDAC1 HDAC4 HDAC6 HDAC11 EZH2^wt^	0.19 µM >10 µM 0.03 µM >10 µM 0.59 nM	[145]
St.37	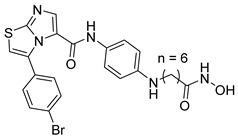	HCT-116 SW480 MDA-MB-231 MCF-7 HK2	16.42 µM 23.43 µM 43.74 µM 36.46 µM >80 µM	HDAC1 HDAC6 IDO1	1.078 µM 58.23 nM 86 nM	[146]
St.38	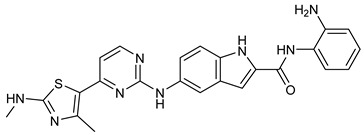	MDA-MB-231 HeLa HepG2	2.47 µM 1.51 µM 4.52 µM	HDAC1 HDAC2 HDAC3 CDK9	1.73 µM >50 µM 1.11 µM 0.17 µM	[147]
St.39	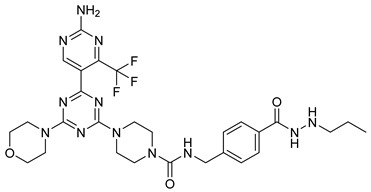	JEKO-1	0.9 µM	HDAC1 HDAC2 HDAC3 PI3K α PI3K β	75.5 nM 70.9 nM 1.9 nM 2.5 nM 10.0 nM	[148]

In a recent study, researchers developed a novel class of tazemetostat-derived HDAC/EZH2 dual inhibitors to exploit the therapeutic synergy of co-targeting epigenetic regulators in hematological malignancies. Among the synthesized compounds, St.36 (Table 5) showed outstanding dual-inhibitory potency, displaying strong antiproliferative activity against EZH2-mutant DLBCL and multiple AML cell lines [145]. In a recent study, a series of imidazothiazole-based hydroxamic acid derivatives was developed as dual IDO1/HDAC6 inhibitors using a pharmacophore fusion strategy. Among them, St.37 (Table 5) emerged as the most potent, exhibiting strong IDO1 inhibition and high HDAC6 selectivity over other isoforms, validated by Western blot. Docking studies suggested favorable binding to both IDO1 and HDAC6. Biologically, St.37 induced G2/M cell cycle arrest in HCT-116 cells and showed significant in vivo antitumor activity in CT26 tumor-bearing mice with minimal toxicity [146].

Several recent studies have focused on the development of dual HDAC/CDK inhibitory agents [147,149]. In this study, a series of dual CDK9/HDACi based on N-(2-aminophenyl)-5-(4-aryl-pyrimidin-2-yl)amino-1H-indole-2-carboxamide scaffolds was designed and synthesized to target transcriptional dysregulation in cancer. Among them, St.38 (Table 5) exhibited strong antiproliferative activity across multiple cancer cell lines and potently inhibited CDK9, HDAC1, and HDAC3. It induced mitochondria-mediated apoptosis and G_2_/M arrest, outperforming reference inhibitors AZD-5438 and Mocetinostat. In vivo, St.38 significantly reduced MDA-MB-231 tumor growth by 76.8%, establishing it as a promising dual CDK9/HDAC inhibitor with potent anticancer potential [147]. In another study, a series of dual PI3K/HDACi was designed by integrating the morpholino-triazine core of the PI3K inhibitor with the hydrazide moiety of an HDAC1–3 selective inhibitor. Among them, St.39 (Table 5) underscored potent inhibition of both PI3K isoforms (IC_50_ = 2.5–80.5 nM) and HDAC1–3 (IC_50_ = 1.9–75.5 nM), exhibiting strong antiproliferative effects across multiple cancer cell lines. In mantle cell lymphoma (Jeko-1) cells, and induced markedly better apoptosis than either single-target inhibitor [148].

## 7. Conclusions

HDACi remain one of the most dynamic and rapidly evolving classes of epigenetic anticancer agents, with several FDA-approved drugs establishing HDACs as clinically validated therapeutic targets. Nevertheless, the majority of clinically advanced HDACi continue to rely on classical hydroxamate or non-hydroximate ZBGs, which, despite their strong zinc-chelating potency, are limited by metabolic instability, suboptimal pharmacokinetics, and dose-limiting toxicities. This review systematically evaluated HDACi reported in recent years, highlighting key innovations across the three essential pharmacophore components—CAP region engineering, linker optimization, and ZBG diversification. Comparative assessment of biological activities across cancer models and HDAC isoforms reveals clear medicinal chemistry trends that are redefining the design of next-generation HDACi. Structural analyses show that tailoring the CAP group and fine-tuning linker rigidity can substantially enhance potency and selectivity, even when employing weaker but safer ZBGs. Growing evidence indicates that strategic ZBG diversification, rather than simply maximizing Zn^2+^ affinity, represents the most promising pathway toward generating HDACi with improved drug-like properties, reduced off-target effects, and greater clinical feasibility.

Importantly, recent advances demonstrate that computational and in silico methodologies—including ligand-based virtual screening, docking studies, molecular dynamics simulations, and predictive pharmacokinetic modeling—offer powerful tools to accelerate HDACi discovery. These approaches provide deep insights into binding interactions, isoform selectivity, ZBG behavior within zinc-containing catalytic pockets, and the overall structural determinants of activity. By integrating these computational techniques with experimental validation, researchers can rapidly prioritize promising scaffolds, refine molecular designs, and more efficiently advance candidates with genuine translational potential.

As the field progresses, the synergy between innovative ZBG chemotypes, isoform-selective CAP motifs, optimized linker architectures, and advanced computational design strategies is poised to deliver the next generation of HDAC inhibitors. These emerging compounds are expected to be not only potent but also safer, more selective, and more clinically viable—marking a significant step forward in the development of durable epigenetic therapies for cancer and other diseases.

## Figures and Tables

**Figure 1 cells-14-01997-f001:**
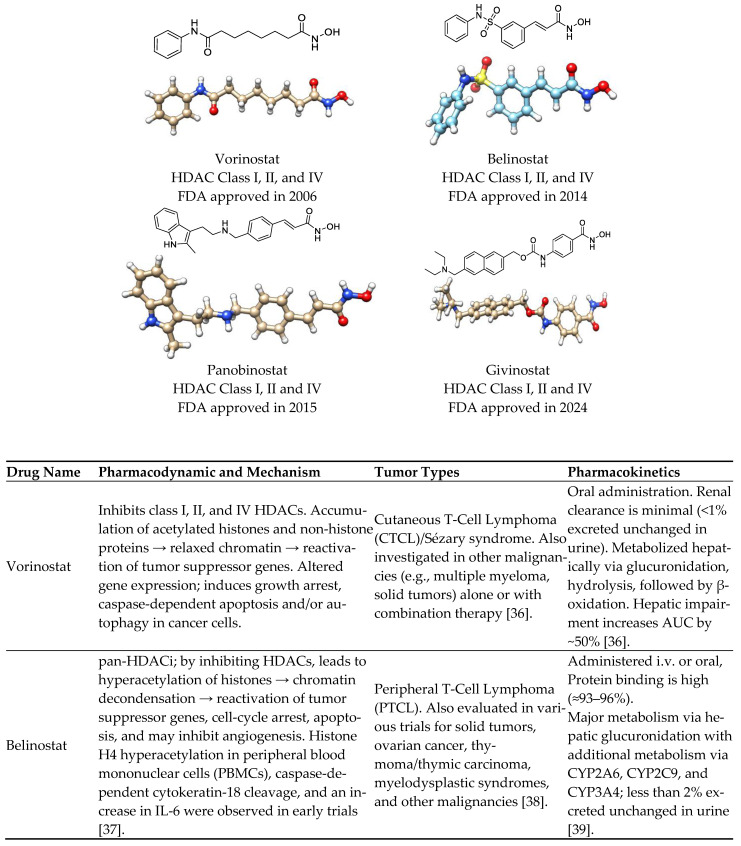
The 3D chemical structure, Pharmacokinetics, Pharmacodynamics, Mechanism of action, Tumor Types, and specificity of HDAC inhibitor approved drugs with hydroximate scaffold [36,37,38,39,40,41,42,43,44].

**Figure 2 cells-14-01997-f002:**
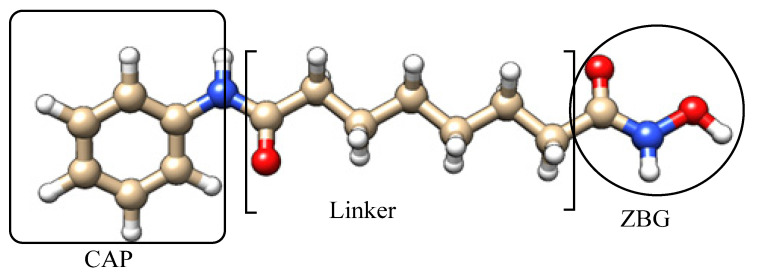
General pharmacophore architecture of HDACi, highlighting the three essential structural domains (blue for Oxygen, red for nitrogen, white for hydrogen and grey for carbon atoms).

**Figure 3 cells-14-01997-f003:**
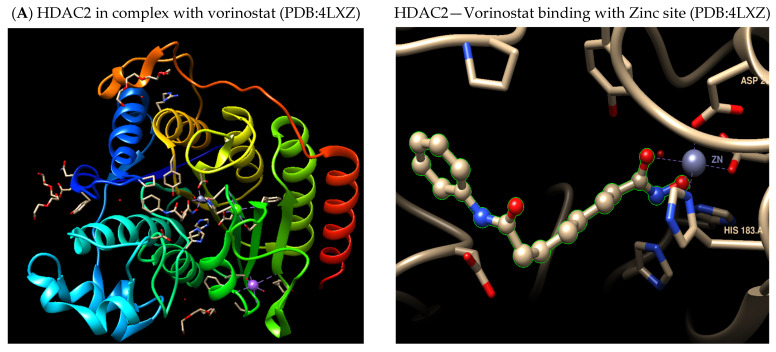
Binding pose of Vorinostat at (**A**) HDAC2 zinc catalytic site based on the experimental crystal structure (PDB: 4LXZ). (**B**) HDAC6 zinc catalytic site based on the experimental crystal structure (PDB: 5EEI). (**C**) HDAC8 zinc catalytic site based on the experimental crystal structure (PDB: 4BZ6). Dashed lines indicate Zn^2+^–ligand coordination interactions.

**Table 2 cells-14-01997-t002:** The structures and the IC_50_ values against a panel of cancer cell lines and HDAC enzymes for the most active agents, which contain hydroximate scaffolds.

Code	Structures	Evaluated Cancer/Normal Cell Lines		Evaluated Target		Ref.
		Cell Lines	IC_50_/%V	HDAC	IC_50_	
St.1	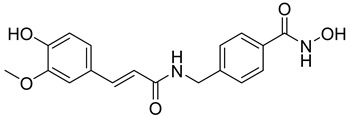	U87-MG T98G U251-MG	51.31 µM 42.60 µM 2.37 µM	HDAC1 HDAC2 HDAC3 HDAC5 HDAC6 HDAC10	403 nM 537 nM 1.278 µM 4.455 µM 4.5 nM 202 nM	[91]
St.2	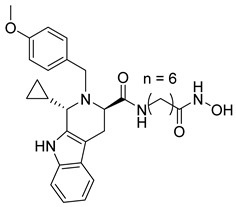	HCT116 SK-MEL-2 HS-5	1.54 µM 0.48 µM >50 µM	HDAC1 HDAC2 HDAC3 HDAC6	8.73 nM 23.5 nM 32.1 nM >50 nM	[92]
St.3	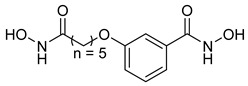	NA	NA	HDAC1	2.96 µM	[93]
St.4	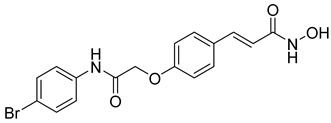	THP-1	1.60 µM	HDAC I/II	157.0 nM	[94]
St.5	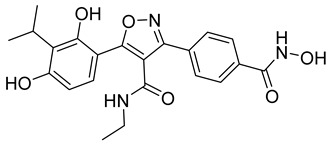	PC3 PDXO	65 nM 345 nM	HDAC1	increased histone H3 acetylation	[95]
St.6	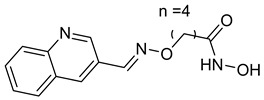	UM	NA	HDAC1 HDAC3 HDAC6 HDAC8	0.137 µM 0.040 µM 0.010 µM 9.29 µM	[96]
St.7	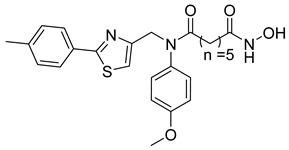	HepG2 PC9 HCT116 MCF-7	0.19 µM 0.46 µM 1.19 µM 3.31 µM	HDAC1 HDAC6	0.8 nM 2.5 nM	[97]
St.8	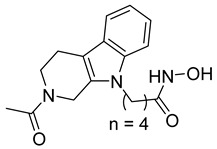	A549	1.09 µM	HDAC1	4.5 nM	[98]
St.9	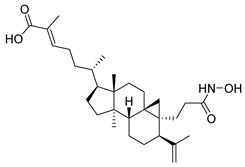	J774A (A) LDH (B) IL-1β	9.98 µM 5.50 µM	HDAC1 HDAC2 HDAC4 HDAC6 HDAC8	1.14 µM 10.56 µM 19.39 µM 2.23 µM >50 µM	[99]
St.10	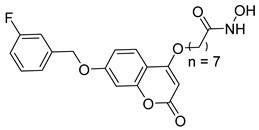	4T1 MDAMB231	21.4% 25.4%	HDAC1	0.99 µM	[100]
St.11	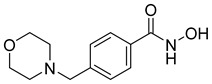	LX-2	NA	HDAC1 HDAC6	5.9 nM 78.1 nM	[89]
St.12	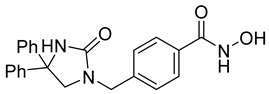	LX-2	NA	HDAC1 HDAC6	8.8 nM 50.8 nM	
St.13	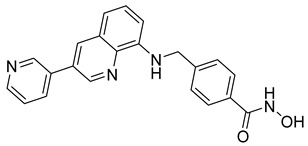	A549 HCT116	1.29 µM 1.61 µM	HDAC1 HDAC2 HDAC6 HDAC8	2.62 µM 1.31 µM 4.75 nM 1.80 µM	[101]
St.14	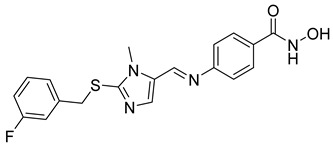	A2780	8.10 µM	Pan-HDAC	12.58 µM	[102]
St.15	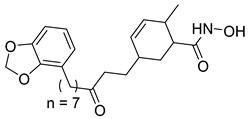	HT-29 MDA-MB-231	4.02 µM 2.31 µM	HDAC8	16.11 nM	[103]
St.16	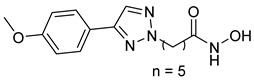	A549 MCF-7	<10 µg/ml	HDAC1 HDAC6	3.06 µM 4.08 µM	[104]
St.17	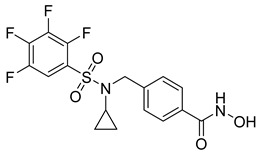	MV4-11 MRC-9	0.42 µM >20 µM	HDAC6 HDAC8	8.50 nM 0.334 µM	[105]
St.18	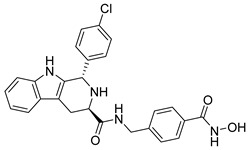	NA	NA	HDAC6	2.68 nM	[106]
St.19	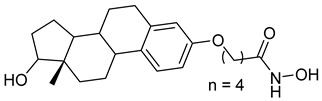	HeLa	6.74 µM	HDAC	6.23 µM	[107]
St.20	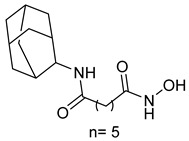	NA	NA	HDAC6	0.96 µM	[108]

NA: Not applicable (no cell lines were evaluated in these works).

**Table 3 cells-14-01997-t003:** Representative Benzamide-Based HDACi in Clinical Trials: Structures, Targets, and Cancer Applications.

Name or Code	Structure	Clinical Trial	Cancer Type	HDAC Class	Ref.
Entinostat	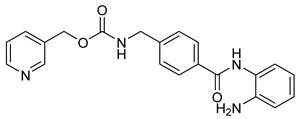	I, II & III	Solid tumors, Chronic Myeloid Leukemia, & Acute Myeloid Leukemia	I	[119,120]
Chidamide	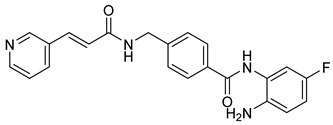	Ib/II/III	T-cell Lymphoma, Angioimmunoblastic T-cell Lymphoma, lymphoma & Breast cancer	HDAC 1,2,3,10	[121,122]
Tacedinaline	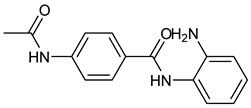	II/III	Advanced pancreatic cancer & Multiple Myeloma	I	[123,124]
Mocetinostat	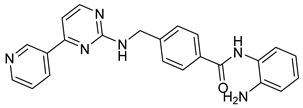	II	classical Hodgkin Lymphoma & Relapsed/Refractory Lymphoma	I & IV	[125]
Zabadinostat (CXD-101)	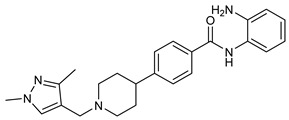	I	solid tumours, lymphoma, and myeloma	I	[126,127]

**Table 4 cells-14-01997-t004:** The structures of Non-Hydroxamate-Based HDACi and the IC50 values against a panel of cancer cell lines and HDAC enzymes for the most active agents.

Code	Structures	Evaluated Cancer/Normal Cell Lines		Evaluated Target		Ref.
		Cell Lines	IC_50_	HDAC	IC_50_	
St.21	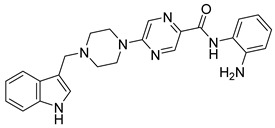	MV4-11 MOLM-13	194 nM 318 nM	HDAC1 HDAC2 HDAC3	0.13 µM 0.28 µM 0.31 µM	[128]
St.22	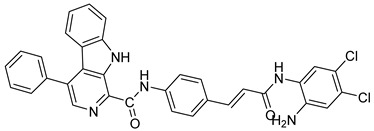	HCT-15 HT-29 L-132	0.70 µM 0.94 µM 14.50 µM	HDAC I	0.97 µM	[129]
St.23	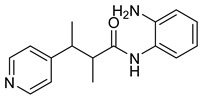	MDA-MB-231	34.3 µM	HDAC3	0.69 µM	[130]
St.24	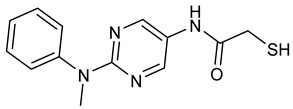	NA	NA	HDAC1 HDAC6	2.124 µM 73 nM	[131]
St.25	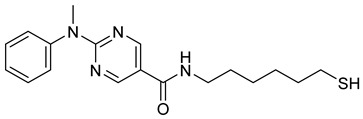	NA	NA	HDAC1 HDAC6	13.0 nM 4.0 nM	[131]
St.26	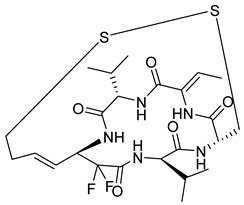	HT-29	<150 nM	HDAC1 HDAC2 HDAC3 HDAC8 HDAC6	0.948 nM 0.856 nM 1.06 nM 4.24 nM >1000 nM	[132]
St.27	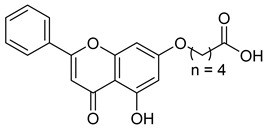	HCT-116	13.04 µM	HDAC8	75.37 µM	[133]
St.28	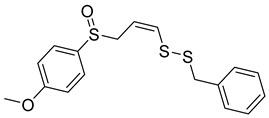	BE(2)-C IMR-32 SH-SY5Y	2.52 µM 1.50 µM 1.72 µM	HDAC6 HDAC8	10.78 µM 0.81 µM	[134]

NA: Not applicable (no cell lines were evaluated in these works).

## Data Availability

This is a Review article, and there are no new data in this manuscript.

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
