# Peer review of "Next-Generation HDAC Inhibitors: Advancing Zinc-Binding Group Design for Enhanced Cancer Therapy"

_cells, 2025, doi:10.3390/cells14241997_

Round 1

Reviewer 1 Report

Comments and Suggestions for Authors

This is a comprehensive review manuscript about the current state and recent developments of HDAC inhibitors for the therapy of cancer diseases. However, I have some concerns about the novelty and quality of the manuscript:

Several review manuscripts about anticancer active HDAC inhibitors were published over the last few years. So it remains unclear to me how far this new manuscript differs from those review papers already published and in parts cited in this submission.

As a stylistic issue, the chemical structures in the tables are inconsistent in their size and need to be improved. Moreover, the style of the references is inconsistent.

Author Response

Dear Respected reviewer,

      Many thanks for providing me with the opportunity to revise my manuscript. I would also like to thank you for your time and expertise in providing feedback.

I think that all of your comments are legitimate and require consideration. I would like to profoundly thank you for your constructive comments that have significantly improved my manuscript.

Please find below my response to your comments point by point. I have considered all of the comments carefully and have amended the manuscript as appropriate. The amended text is highlighted throughout the manuscript. I have provided a detailed response to each of the comments.

Response to Reviewer 1 Comments

Comment 1: This is a comprehensive review manuscript about the current state and recent developments of HDAC inhibitors for the therapy of cancer diseases. However, I have some concerns about the novelty and quality of the manuscript:

Several review manuscripts about anticancer active HDAC inhibitors were published over the last few years. So it remains unclear to me how far this new manuscript differs from those review papers already published and in parts cited in this submission.

Response 1: I sincerely thank the reviewer for this valuable and constructive comment. The proposed review manuscript differentiates itself from existing literature on HDAC inhibitors (HDACi) by providing a critically focused and ultra-current analysis of the field's shift towards mitigating the inherent liabilities of FDA-approved agents. While prior reviews covered general concepts, this work focuses specifically on the development of hydroxamate and non-hydroxamate inhibitors—such as benzamides, hydrazides, and thiols—to address the metabolic instability, genotoxicity, and poor pharmacokinetics associated with conventional zinc-binding groups (ZBGs). Crucially, the manuscript provides an in-depth Structure-Activity Relationship (SAR) analysis of the cap, linker, and ZBG motifs for both novel hydroxamate- and non-hydroxamate-based compounds, a level of chemical detail often lacking in broader reviews. Furthermore, its novelty is cemented by the inclusion of clinical trial ID numbers and phases for relevant compounds, and by systematically evaluating emerging dual-target HDAC inhibitors (e.g., HDAC–PI3K and HDAC–CDK9 hybrids), synthesizing approximately 75 recent works from the current year (2025) to ensure the most timely and progressive overview of therapeutic potential, potency, and isoform selectivity profiles.

Comment 2: As a stylistic issue, the chemical structures in the tables are inconsistent in their size and need to be improved. Moreover, the style of the references is inconsistent.

 Response 2: All structures were checked and drawn with the same size rationally by using ACS document style, as well as all references were controlled and edited accordingly.

Best Regards

Reviewer 2 Report

Comments and Suggestions for Authors

This review article provides a comprehensive analysis of recently developed HDAC inhibitors reported in the last few years, emphasizing their structure–activity relationships (SAR), chemical scaffolds, and binding features—including cap, linker, and ZBG motifs. The main critique of this manuscript is that it lists all approaches and does not critically address reported work. A review article should be more than just a summary of literature. As this presented work in mainly the summary of literature is also quite demanding reading. Someone might say it is “dry” and does not bring special interest for a reader. To often there are presented paragraphs without any critical input. I suggest to reorganize the manuscript taking in comsideration the following comments:

a) Introduction line 63. Please insert a reference related to the hydrazide group’s role in HDAC inhibitor development, highlighting their pharmaceutical properties, biological activities, and potential benefits in reducing side effects (Raucci et al. J. Med. Chem. 2025, 68, 14171−14194).

b) In the table 1, titled “Representative Hydroxamate-Based HDACi in Clinical Trials: Structures, Targets, and Cancer Applications” for each compound I suggest to indicate company’s name, status and clinical trial ID.

c) In the list of Hydroxamate-Based HDACi (line 171) several compoyunds in clinical trials such as REC-2282, Ivaltinostat, Bisthianostat and R-306465 were not cited.

d) A chapter on more recent HDAC-based dual-target inhibitors,should be included, such as: Xie, S.; et al, . Design and biological evaluation of dual tubulin/HDAC inhibitors based on millepachine for treatment of prostate cancer. Eur. J. Med. Chem. 2024, 268, 116301.

e) Line 423, Table 4. Representative Benzamide-Based HDACi in Clinical Trials: Structures, Targets, and Cancer Applications. Please insert CDX-101

f) The recent progress on HDAC-based drug discovery with a focus on HDAC inhibitor-based drug combination therapy and HDAC-targeting strategies such as PROTAC HDAC degraders should be also included.

Author Response

Dear Respected reviewer,

      Many thanks for providing me with the opportunity to revise my manuscript. I would also like to thank you for your time and expertise in providing feedback.

I think that all of your comments are legitimate and require consideration. I would like to profoundly thank you for your constructive comments that have significantly improved my manuscript.

Please find below my response to your comments point by point. I have considered all of the comments carefully and have amended the manuscript as appropriate. The amended text is highlighted throughout the manuscript. I have provided a detailed response to each of the comments.

Response to Reviewer 2 Comments

This review article provides a comprehensive analysis of recently developed HDAC inhibitors reported in the last few years, emphasizing their structure–activity relationships (SAR), chemical scaffolds, and binding features—including cap, linker, and ZBG motifs. The main critique of this manuscript is that it lists all approaches and does not critically address reported work. A review article should be more than just a summary of literature. As this presented work in mainly the summary of literature is also quite demanding reading. Someone might say it is “dry” and does not bring special interest for a reader. To often there are presented paragraphs without any critical input. I suggest to reorganize the manuscript taking in comsideration the following comments:

I sincerely thank the reviewer for this valuable and constructive comment. I fully agree that a high-quality review article must go beyond summarizing previously published work and should instead provide critical interpretation, highlight challenges, and synthesize broader insights. In response, I have substantially revised the manuscript to enhance its analytical depth and improve readability. Throughout the SAR discussion, particularly in the CAP, linker, and ZBG sections, I have added critical evaluation to explain why certain structural modifications influence potency, selectivity, pharmacokinetics, or toxicity, and we now comment on conflicting findings or limitations in the reported studies. I believe these revisions address the reviewer’s concerns and transform the manuscript into a more critical, insightful, and reader-oriented review.

Comment a) Introduction line 63. Please insert a reference related to the hydrazide group’s role in HDAC inhibitor development, highlighting their pharmaceutical properties, biological activities, and potential benefits in reducing side effects (Raucci et al. J. Med. Chem. 2025, 68, 14171−14194).

Response a: Thank you for this helpful suggestion. I have been now revised the introduction to incorporate the recommended reference (Raucci et al., J. Med. Chem. 2025, 68, 14171–14194) and expanded our discussion of the hydrazide group. Specifically, it was added a detailed explanation of the emerging importance of hydrazides as alternative zinc-binding groups (ZBGs), emphasizing their favorable pharmaceutical properties, biological activities, and potential to reduce toxicity compared to traditional hydroxamic acids and 2-aminoanilides.

Comment b) In the table 1, titled “Representative Hydroxamate-Based HDACi in Clinical Trials: Structures, Targets, and Cancer Applications” for each compound I suggest to indicate company’s name, status and clinical trial ID.

Response b: I would like to thank the reviewer for this valuable comment. In response, Table 1 has been updated to include—where available—the company or developer associated with each compound, its most recent clinical development status, and the corresponding ClinicalTrials.gov Identifier (NCT number). These additions improve the clarity, completeness, and utility of the table, allowing readers to better understand the translational progress and industry involvement behind each HDAC inhibitor currently in clinical evaluation. All new information has been incorporated into the revised table, with references adjusted accordingly.

Comment c) In the list of Hydroxamate-Based HDACi (line 171) several compoyunds in clinical trials such as REC-2282, Ivaltinostat, Bisthianostat and R-306465 were not cited.

Response c: thank you for this valuable comment too, the mentioned compounds were added to the paragraph and table 1, and all of the compounds’ SAR were discussed accordingly

Comment d) A chapter on more recent HDAC-based dual-target inhibitors,should be included, such as: Xie, S.; et al, . Design and biological evaluation of dual tubulin/HDAC inhibitors based on millepachine for treatment of prostate cancer. Eur. J. Med. Chem. 2024, 268, 116301.

Response d: a separate chapter regards recent HDAC-based dual-target inhibitors were added to this review work including the suggested work and other recent works.

Comment e) Line 423, Table 4. Representative Benzamide-Based HDACi in Clinical Trials: Structures, Targets, and Cancer Applications. Please insert CDX-101

Response e: Thank you for this suggestion too, but to be clear you mean CXD-101 not CDX-101, since CDX-101 is not HDAC inhibitor, However, CXD-101 (Zabadinostat) was added to the mentioned table accordingly, the table is 3 now after the editing according to the other reviewer’s comment

Comment f) The recent progress on HDAC-based drug discovery with a focus on HDAC inhibitor-based drug combination therapy and HDAC-targeting strategies such as PROTAC HDAC degraders should be also included.

Response f: I thank the reviewer for this insightful comment. A new paragraph has now been added to the Introduction to briefly highlight recent progress in HDAC-based drug discovery, including HDAC inhibitor combination therapies and emerging HDAC-targeting strategies such as PROTAC degraders. This addition provides appropriate contextualization of current trends in the field. However, as the primary focus of this review is the medicinal chemistry and SAR-driven development of recently reported small-molecule HDAC inhibitors—particularly innovations in ZBG design—rather than combination therapy or protein degradation strategies.

Best Regards

Reviewer 3 Report

Comments and Suggestions for Authors

Specific Comments to the Authors

The submitted review, “Next-Generation HDAC Inhibitors: Advancing Zinc-Binding Group Design for Enhanced Cancer Therapy”, provides a comprehensive and up-to-date overview of recently developed HDAC inhibitors (HDACis). It effectively highlights their chemical and pharmacological properties in relation to in vitro, in vivo, and clinical findings.

The topics covered span from classical chemical and pharmaceutical characteristics of HDACis to mechanistic aspects observed in vitro, in vivo, and in situ, as well as current and future strategies for the development of next-generation HDACis.

In summary, the review offers valuable insights into the potential impact and promising opportunities of newly developed HDAC inhibitors in epigenetic-based targeted therapy. However, the manuscript is at times difficult to read and follow. To enhance clarity and coherence, the structure should be reorganized to establish a stronger narrative flow, with greater emphasis placed on the significance of HDACi development.

Major Concerns

After multiple readings of the manuscript, it is clear that the overall readability requires substantial improvement. The authors should explicitly highlight which HDAC inhibitors (HDACis) are genuinely novel and identify those with the greatest potential for translation from experimental studies to clinical applications. In addition, the chemical design of dual-target HDACis should be presented and discussed in a dedicated section, as such dual-drug development represents an emerging and highly promising strategy in cancer therapy (see for example FASEB J. 2025 Oct 31;39(20):e71152. doi: 10.1096/fj.202502614R. PMID: 41105425).

Minor Concerns

# Title: The title refers to “Advancing Zinc-Binding Group Design for Enhanced Cancer Therapy.” However, the manuscript should more clearly explain how this chemical design influences the anticancer properties of next-generation HDACis.

# Abstract: The statement regarding “emerging dual-target HDAC inhibitors, such as HDAC–PI3K and HDAC–CDK9 hybrids, discussed for their synergistic anticancer effects” should be elaborated further, ideally in a separate chapter.

# Figure 1: Figure 1 should be supplemented with an additional table containing basic pharmaceutical data (pharmacokinetics and pharmacodynamics) and clinical information (tumor type, stage of therapy, etc.), as well as their pathological mechanisms in relation to the hallmarks of cancer for the FDA-approved HDAC inhibitors mentioned.

# Figure 3: Please clarify the readout for the presented binding site in relation to docking and molecular dynamics.

# Table 1: Add key pharmaceutical data for the FDA-approved HDACis, along with relevant clinical information on biosafety, genotoxicity, and major endpoints (e.g., OS, DFS, PFS).

# Table 2: Indicate which of these active agents are currently under clinical investigation, and add their primary mechanisms of action.

# Table 3: Explain why this table is separate from Table 2, given that the table legends are identical.

# Tables 4 and 5: Clearly indicate the differences between these tables in their legends.

# Conclusions: Please include comments on how chemical in silico investigations can contribute to the development of new or next-generation HDAC inhibitors. Such computational approaches may provide valuable insights into molecular design, binding interactions, and predictive pharmacological properties, thereby accelerating the identification of promising candidates for clinical translation. For reference, see Chem Biodivers. 2025 Sep 2:e01492. doi: 10.1002/cbdv.202501492 (PMID: 40893045) and Comput Biol Med. 2025 Sep;196(Pt A):110695. doi: 10.1016/j.compbiomed.2025.110695 (PMID: 40617084).

Author Response

Dear Respected reviewer,

      Many thanks for providing me with the opportunity to revise my manuscript. I would also like to thank you for your time and expertise in providing feedback.

I think that all of your comments are legitimate and require consideration. I would like to profoundly thank you for your constructive comments that have significantly improved my manuscript.

Please find below my response to your comments point by point. I have considered all of the comments carefully and have amended the manuscript as appropriate. The amended text is highlighted throughout the manuscript. I have provided a detailed response to each of the comments.

Response to Reviewer 3 Comments

The submitted review, “Next-Generation HDAC Inhibitors: Advancing Zinc-Binding Group Design for Enhanced Cancer Therapy”, provides a comprehensive and up-to-date overview of recently developed HDAC inhibitors (HDACis). It effectively highlights their chemical and pharmacological properties in relation to in vitro, in vivo, and clinical findings.

The topics covered span from classical chemical and pharmaceutical characteristics of HDACis to mechanistic aspects observed in vitro, in vivo, and in situ, as well as current and future strategies for the development of next-generation HDACis.

In summary, the review offers valuable insights into the potential impact and promising opportunities of newly developed HDAC inhibitors in epigenetic-based targeted therapy. However, the manuscript is at times difficult to read and follow. To enhance clarity and coherence, the structure should be reorganized to establish a stronger narrative flow, with greater emphasis placed on the significance of HDACi development.

 Response : Many thanks for the reviewer’s valuable feedback, which has substantially contributed to improving the quality and clarity of the manuscript.

Major Concerns

Comments 1: After multiple readings of the manuscript, it is clear that the overall readability requires substantial improvement. The authors should explicitly highlight which HDAC inhibitors (HDACis) are genuinely novel and identify those with the greatest potential for translation from experimental studies to clinical applications. In addition, the chemical design of dual-target HDACis should be presented and discussed in a dedicated section, as such dual-drug development represents an emerging and highly promising strategy in cancer therapy (see for example FASEB J. 2025 Oct 31;39(20):e71152. doi: 10.1096/fj.202502614R. PMID: 41105425).

Response 1: I would like to sincerely thank the reviewer for this important comment. To improve the overall readability and clarity of the manuscript, It was substantially revised the introduction and aim sections. As well as almost all of the compounds in this work are novel compounds in their recent works and they have a potential for translation into future clinical applications, based on their most recent preclinical performance. In addition, and in accordance with the reviewer’s recommendation, a dedicated section discussing the chemical design strategies and recent developments in dual-target HDAC inhibitors was added to this work.

Minor Concerns

Comment 2#: Title: The title refers to “Advancing Zinc-Binding Group Design for Enhanced Cancer Therapy.” However, the manuscript should more clearly explain how this chemical design influences the anticancer properties of next-generation HDACis.

Response2: I appreciate the reviewer’s insightful observation regarding the need to more clearly explain how zinc-binding group (ZBG) design influences the anticancer properties of next-generation HDAC inhibitors. In response, I have added a new paragraph in the Introduction that explicitly discusses the impact of optimized ZBG architecture on binding affinity, isoform selectivity, metabolic stability, and overall therapeutic efficacy. This addition clarifies the mechanistic relevance of ZBG innovations and strengthens the alignment between the manuscript’s title and its scientific content.

Comment 3#: Abstract: The statement regarding “emerging dual-target HDAC inhibitors, such as HDAC–PI3K and HDAC–CDK9 hybrids, discussed for their synergistic anticancer effects” should be elaborated further, ideally in a separate chapter.

Response 3: a separate chapter regards recent HDAC-based dual-target inhibitors were added to this review work including recent works.

Commnet 4#: Figure 1: Figure 1 should be supplemented with an additional table containing basic pharmaceutical data (pharmacokinetics and pharmacodynamics) and clinical information (tumor type, stage of therapy, etc.), as well as their pathological mechanisms in relation to the hallmarks of cancer for the FDA-approved HDAC inhibitors mentioned.

Response 4: all requested data were collected and added in this figure accordingly.

Comment 5 #: Figure 3: Please clarify the readout for the presented binding site in relation to docking and molecular dynamics.

Response 5: The data presented in this figure have been updated and reorganized for improved clarity. In addition, the figure caption has been revised to clearly explain the readout and its relevance to the docking analyses.

Comment 6#: Table 1: Add key pharmaceutical data for the FDA-approved HDACis, along with relevant clinical information on biosafety, genotoxicity, and major endpoints (e.g., OS, DFS, PFS).

Response 6: Table 1 does not originally contain FDA-approved HDAC inhibitors; however, the table has been updated in accordance with your valuable comment, as well as the suggestions provided by the other reviewer.

Comment 7#: Table 2: Indicate which of these active agents are currently under clinical investigation, and add their primary mechanisms of action.

Response 7: Thank you for this valuable comment. The mechanisms of action have been added to Table 2, identifying these compounds as HDAC inhibitors together with their corresponding IC₅₀ values. However, it should be noted that most of these agents were extracted from recent studies and have not yet advanced into clinical trials.

Comment 8#: Table 3: Explain why this table is separate from Table 2, given that the table legends are identical.

Response 8: these both tables were incorporated in one table now (Table 2)

Comment 9 #: Tables 4 and 5: Clearly indicate the differences between these tables in their legends.

Response 9: Following the revisions, the referenced tables now appear as Tables 3 and 4. Table 3 presents benzamide-based HDAC inhibitors currently in clinical trials, while Table 4 summarizes recently reported non-hydroxamate HDAC inhibitors that exhibit potent activity and may hold promise for future clinical evaluation.

Comment 10# :Conclusions: Please include comments on how chemical in silico investigations can contribute to the development of new or next-generation HDAC inhibitors. Such computational approaches may provide valuable insights into molecular design, binding interactions, and predictive pharmacological properties, thereby accelerating the identification of promising candidates for clinical translation. For reference, see Chem Biodivers. 2025 Sep 2:e01492. doi: 10.1002/cbdv.202501492 (PMID: 40893045) and Comput Biol Med. 2025 Sep;196(Pt A):110695. doi: 10.1016/j.compbiomed.2025.110695 (PMID: 40617084).

Response 10: I would like to thank the reviewer for this insightful comment emphasizing the importance of in silico investigations in the development of next-generation HDAC inhibitors. In accordance with the reviewer’s recommendation, Section 3 has been updated to include detailed examples of how computational analyses elucidate zinc chelation mechanisms and guide the rational design of new scaffolds and alternative ZBG chemotypes. Additionally, the Conclusion section has been comprehensively revised to reflect the growing significance of computational strategies in HDAC inhibitor discovery and their role in supporting safer, more selective, and more translatable HDACi development.

Best Regards

Round 2

Reviewer 1 Report

Comments and Suggestions for Authors

The revised manuscript is suitable for publication now.

Author Response

I would like to thank the reviewer for his positive feedback.

Reviewer 2 Report

Comments and Suggestions for Authors

The review article has been significantly improved and it is acceptable for publication in the actual form. 

Author Response

I would like to thank the reviewer for his positive feedback

Reviewer 3 Report

Comments and Suggestions for Authors

Specific comments to the authors

In the revised version of the manuscript, the authors have addressed the previously raised concerns in a clear, adequate, and convincing manner. For final acceptance, the following minor revisions should be implemented: the legend of Table 5 should explicitly identify the compounds as HDAC-based dual-target inhibitors, consistent with the discussion in Chapter 6 (“HDAC-based dual-target inhibitors”). In addition, the thoroughly revised manuscript should be carefully checked for typographical errors (e.g., “HDACI” on page 21, line 572). With these minor corrections, the manuscript “Next-Generation HDAC Inhibitors: Advancing Zinc-Binding Group Design for Enhanced Cancer Therapy” will be suitable for publication in “Cells”.

Author Response

Dear Respected reviewer,

      Many thanks for providing me with the opportunity to revise my manuscript again. I would also like to thank you for your time and expertise in providing feedback.

Please find below my response to your comments point by point. I have considered all of the comments carefully and have amended the manuscript as appropriate. The amended text is highlighted throughout the manuscript. I have provided a detailed response to each of the comments.

Response to Reviewer 3 Comments

Comment 1: In the revised version of the manuscript, the authors have addressed the previously raised concerns in a clear, adequate, and convincing manner.

Response 1: I sincerely thank the reviewer for this valuable and constructive comment

Comment 2: For final acceptance, the following minor revisions should be implemented: the legend of Table 5 should explicitly identify the compounds as HDAC-based dual-target inhibitors, consistent with the discussion in Chapter 6 (“HDAC-based dual-target inhibitors”).

Response 2: Thank you for this comment and note, the legend of this table edited now accordingly

Comment 3: In addition, the thoroughly revised manuscript should be carefully checked for typographical errors (e.g., “HDACI” on page 21, line 572). With these minor corrections, the manuscript “Next-Generation HDAC Inhibitors: Advancing Zinc-Binding Group Design for Enhanced Cancer Therapy” will be suitable for publication in “Cells”.

Response 3: the whole manuscript was checked again for all typography and grammar errors accordingly.

Best Regards